# Impact of Landscape on Host–Parasite Genetic Diversity and Distribution Using the *Puumala orthohantavirus*–Bank Vole System

**DOI:** 10.3390/microorganisms9071516

**Published:** 2021-07-15

**Authors:** Maria Razzauti, Guillaume Castel, Jean-François Cosson

**Affiliations:** 1CBGP, INRAE, CIRAD, IRD, Montpellier SupAgro, Université Montpellier, 34000 Montpellier, France; guillaume.castel@inrae.fr; 2UMR BIPAR, Animal Health Laboratory, ANSES, INRAE, Ecole Nationale Vétérinaire d’Alfort, Université Paris-Est, 94700 Maisons-Alfort, France; jean-francois.cosson@inra.fr

**Keywords:** zoonoses, *Puumala orthohantavirus*, landscape genetics, genetic diversity, virus distribution, parasite–host interaction, population dynamics, epidemiology, host–parasite

## Abstract

In nature, host specificity has a strong impact on the parasite’s distribution, prevalence, and genetic diversity. The host’s population dynamics is expected to shape the distribution of host-specific parasites. In turn, the parasite’s genetic structure is predicted to mirror that of the host. Here, we study the tandem *Puumala orthohantavirus* (PUUV)–bank vole system. The genetic diversity of 310 bank voles and 33 PUUV isolates from 10 characterized localities of Northeast France was assessed. Our findings show that the genetic diversity of both PUUV and voles, was positively correlated with forest coverage and contiguity of habitats. While the genetic diversity of voles was weakly structured in space, that of PUUV was found to be strongly structured, suggesting that the dispersion of voles was not sufficient to ensure a broad PUUV dissemination. Genetic diversity of PUUV was mainly shaped by purifying selection. Genetic drift and extinction events were better reflected than local adaptation of PUUV. These contrasting patterns of microevolution have important consequences for the understanding of PUUV distribution and epidemiology.

## 1. Introduction

Throughout history, infectious diseases have caused devastating illnesses in humans and livestock. Emerging infections still pose a significant threat to public health and global economy today. Evidently, zoonotic infection emergence is driven by number of socioeconomic and environmental factors [1], e.g., changes in ecosystems that perturb the balance between pathogen and host species, increment of urbanization, increasing long-distance mobility, and trade [2,3,4,5]. Many emerging viruses harbor genomes capable of rapid mutation and selection of new variants in response to environmental changes of hosts and susceptible target species [6,7,8]. Vector-borne and zoonotic diseases often display complex spatial patterns simply because favorable habitats for hosts are linked to environmental factors that vary geographically [9]. Parasites, especially microparasites, directly transmitted between hosts, are indirectly affected by environmental factors influencing their hosts [10]. This phenomenon is even more pronounced when the parasite is host-specific (involving only one host species) and merely asymptomatic, i.e., causing low impact on host survival, population dynamics, and/or host selection. In such cases, the genetic structure and the evolution of parasites are expected to mirror those of its host [11]. Hence, in-depth knowledge of both host–parasite genetics and environmental factors is essential to understand and predict patterns of emergence, spread, and disease control. In this line, the field of landscape genetics is an effective approach to identify relationships between landscape features and genetic variation in free-living organisms [12,13]. It has become a popular method for investigating drivers of processes such as gene flow, genetic drift, and selection [14]. Therefore, a landscape genetics perspective is crucial when elucidating mechanisms underlying basic ecological and evolutionary processes driving infectious disease dynamics and epidemiology. Furthermore, a landscape genetics angle is increasingly used to understand the correlation between spatially dependent population processes and the structural distribution of genetic variation within both hosts and parasites. The growing access to genetic information on hosts and parasites combined with their ecological interactions leads to prediction of the outcomes of host–parasite interactions in natural populations, in terms of both population dynamics and evolutionary interactions [15,16]. Both landscape and environmental features are likely to shape the genetic structure of host populations, and this, in turn, shapes the genetic variability of the virus. A landscape genetics perspective is more and more used to unveil the relationship between the spatial-dependent population processes and the geographic distribution of genetic variation of both hosts and parasites [15].

Hantaviruses are among the main emerging zoonotic viruses in Europe and they represent a serious human health threat worldwide [17,18]. In Eurasia, hantavirus disease or hemorrhagic fever with renal syndrome (HFRS) is an endemic zoonosis that affects tens of thousands of individuals annually [19,20]. The causative agents of hantavirus diseases are viruses of the genus *Orthohantavirus*, family *Hantaviridae* [21]. Many species of rodents, insectivores, and bats act as the host reservoir of hantaviruses [22]. Hantaviruses are generally transmitted between conspecific individuals via aerosolized contaminated excreta and/or bites [23,24,25]. Each orthohantavirus is harbored by a specific rodent species sharing a long-standing virus–host relationship [22,26,27]. Nevertheless, this concept of codivergence has been challenged by the opposing concept of preferential host switching and local host-specific adaptation [28].

This study focuses on *Puumala* virus (PUUV), an orthohantavirus present throughout Europe. When infecting humans, it causes a mild form of HFRS, called nephropathia epidemica (NE). The incidence of NE presents a strong spatial variation through European continent, and this is also evident at a smaller geographical scale [19,29,30,31,32]. Generally, in endemic areas, the fluctuation of NE occurrence in humans is related to the population density of the PUUV host, the bank vole, which in turn is certainly influenced by environmental factors [33,34,35,36]. The bank vole (*Myodes glareolus*) is a rodent belonging to the Arvicolinae subfamily from the Cricetidae family [37]. The bank vole has a Palearctic distribution occupying a broad area of Eurasia from the west coast to east of the Urals, excluding some northern regions and the southern Mediterranean coast, and inhabiting all kinds of moist woodland, preferring densely vegetated clearings, woodland edges, and river banks in forests [38]. The bank voles are unevenly distributed throughout the woodlands, as the nature of landscape structure, such as patch size, fragmentation, and isolation, influences the suitability of its habitats, its dispersal, and the chance for metapopulations to survive [39,40,41,42]. PUUV phylogeny has a strong geographical structuration according to the geographical distribution of its bank vole host populations [43,44,45,46,47,48,49]. The prevalent explanation for the geographical distribution of the eight known PUUV lineages is based on isolation of rodent populations during the Last Glacial Maximum (Weichselian: 10–13 kya) and subsequent recolonization of Eurasia [50,51]. Nevertheless, a less pronounced genetic variation of PUUV is observed at a local scale, generated by genetic drift [49,50,52] and reassortment of genome segments [46,51], while the vast majority of observed point mutations are silent, reflecting stabilizing selection at the amino-acid level [49,52]. Genetic variability of PUUV genomes circulating in one host population might vary from 0% to 10.1%, while this range might be greater when several genogroups or genetic lineages are cocirculating [53]. Nucleotide variability of PUUV between lineages may be as high as 38% for the S segment, and sequence heterogeneity is unevenly distributed along the genome. Nucleotide diversity among orthohantavirus species is variable and probably influenced by the number of strains recognized [53].

The strong host specificity of orthohantaviruses converts their host ecology into a determinant factor that impacts the virus population dynamics and distribution [54]. In this way, the natural and anthropogenic changes of the environment clearly affect the host population, possibly having an impact on the virus population and, indirectly, on the epidemiology of hantaviral diseases in humans [34,35,55]. Therefore, environmental factors, including habitat conditions, food availability, and climate, influence the dynamics and distribution of the bank vole population and, in turn, modulate the incidence of PUUV in this population. The nature and extent of these effects depend on geographical and temporal scales [35]. Moreover, intrinsic factors, such as bank vole behavior [56], movements [57,58], or genetics [49,59,60], affect PUUV transmission among rodents, with consequences for incidence of human cases. Lastly, climate and/or soil composition may affect virus survival outside the host [61,62]. Therefore, understanding the incidence of PUUV infections in a bank vole population is essential for predicting the NE epidemiological risk in humans [31]. 

Several epidemiological studies of infectious diseases included landscape feature data [63,64], whereas others analyzed spatial dynamics of diseases related to land-surface attributes, land cover, and land use [64,65,66]. Only few studies on rodent-borne diseases considered the potential impact of landscape configuration on infectious disease incidence (e.g., [32,33,40,67,68,69,70]). Moreover, despite the growing database of genome sequences and the increasing number of studies on genetic variability of orthohantaviruses, very few have evaluated the influence of the landscape factors and host genetic structure on the viral genetic diversity (e.g., [49]). Overall, the knowledge on how orthohantaviruses and host populations experience and genetically respond to the same landscape phenomena is rarely described in the literature [71,72]; therefore, extended microevolutionary studies of host–parasite models are needed. 

In our previous study, we analyzed the influence of landscape features on dynamics and dispersal of PUUV-infected rodent populations in Northern France [73]. As a follow-up, we aim to understand the link among landscape features, host genetic structuration, and PUUV genetic diversity and circulation. To consider all the important factors when evaluating epidemiological NE risks, we explored the relationship among environmental variables, host abundance and genetics, virus prevalence in host metapopulations, and PUUV diversity. We hypothesize that the land-surface attributes influence the abundance and spatial distribution of bank voles, shaping their population genetics. Hence, landscape would also have an impact on the PUUV prevalence, distribution, and genetic variability.

## 2. Materials and Methods

### 2.1. Samples Preparation

A total of 310 bank voles were trapped in 10 trapping sites in an old forested area of the Northeast part of France, the French Ardennes (Figure 1), during 2008 (a bank vole population density peak year) and 2010. Details of the bank vole (*Myodes glareolus*) sampling procedures are described in our earlier study [73]. Briefly, the voles were annotated (weight, sex, maturity, and age), dissected, and immediately frozen. In order to prevent cross-contamination during dissection, several sets of dissecting instruments were systematically alternated. After dissecting a rodent and harvesting the distinct organs, the set of dissecting instruments used was soaked in Umonium38 (Laboratoire Huckert’s International, Belgium) for 10 min, before rinsing with water and alcohol; the next rodent was dissected with another set. Tissue samples were individually stored in RNAlater (Sigma-Aldrich, St. Louis, MO, USA) and deep-frozen. For this study, a piece of lung (approximately 20 mg each) from each bank vole was collected for analyses.

### 2.2. Microsatellite Genotyping of Voles

A total of 310 bank vole samples were genotyped at 19 unlinked microsatellite loci as described in our previous study [73]. We obtained complete genotypes for 291 individuals at 16 microsatellite loci. Details about procedures and analysis processes can be found in Guivier et al. (2011) [73].

### 2.3. Landscape Characteristics of the Study Sites

Since bank voles are essentially tied to their forested habitat, each study site was characterized by several landscape indices according to few already standardized criteria [74]. The aforementioned criteria were the proportion of forested habitat (*Forest*), the forest contiguity index (*Contig*), the forest shape index (*Shape*), the edge density, the number of patches of forest habitats, and the distance to the next patch. Forest coverage and fragmentation were calculated using ArcGIS^®^ software (Esri. ArcGIS^®^, Redlands, CA, USA) on the basis of the CORINE Land Cover 2012 database. We used a circular buffer of 2 km diameter, which suited both our sampling scheme [73] and what is known about dispersal distances for the bank vole [75].

### 2.4. Screening for PUUV

Rodents were first screened for the presence of PUUV antibodies. Briefly, 50 μL of fresh blood was pipetted from the heart or the thoracic cavity of bank voles. Blood samples were mounted on paper strips (a piece of Whatman blotting paper, 3 × 5 cm), air-dried, and then stored in self-sealing plastic bags at room temperature. Later, blood spots from the paper strips were cut out and eluted in 1 mL of phosphate-buffered saline (PBS). The diluted blood sample of each animal was screened for PUUV immunoglobulin antibodies using the indirect fluorescent antibody test (IFAT). In brief, the PUUV Sotkamo strain-infected Vero E6 cells were detached with trypsin, mixed (in 1:2 ratio) with uninfected Vero E6 cells, washed with PBS, and air-dried on slide spots. Afterward, the slides were fixed with acetone and stored dry at −70 °C until used. Then, 20 µL of each eluted blood sample was added to slides and incubated in a moist chamber at 37 °C for 30 min. Then, the slides were washed three times with PBS and once with distilled water, before incubating at 37 °C for 30 min with FITC anti-mouse polyclonal conjugate (Dako A/S, Copenhagen, Denmark) diluted 1:30 in PBS. Again, the slides were washed three times with PBS and once with distilled water. The prepared slides were studied using a fluorescence microscope. Scattered, granular fluorescence in the cytoplasm of infected Vero E6 cells was considered a positive reaction.

### 2.5. RT-PCR and Sequencing of the PUUV Genome

All samples positive for PUUV-specific antibodies detected by IFAT assay were later analyzed by RT-PCR. Viral RNA from PUUV seropositive voles was extracted from lung tissue samples of PUUV-positive bank voles using the TRIzol reagent (Thermo Fisher Scientific, Waltham, MA, USA) according to the manufacturer’s instructions. Reverse transcription was performed with RevertAid™ H Minus M-MuLV Reverse Transcriptase (Thermo Fisher Scientific, Waltham, MA, USA) to obtain viral cDNA. The complete coding region of the S segment (1–1299 nt) and portions of the M segment (2180–2632 nt) and the L segment (577–987 nt) were amplified using the Multiplex PCR Kit (Qiagen, Hilden, Germany). PCR amplicons were purified with QIAquick PCR Purification Kit (Qiagen, Hilden, Germany). Automated sequencing was performed using the ABI PRISM™ Dye Terminator sequencing kit (Thermo Fisher Scientific, Waltham, MA, USA). To ensure that any possible contamination did not perturb our results, the PUUV RNA extraction–amplification process was done twice, in November 2012 and in April 2013, allowing us to contrast the amplified genome sequences. The newly recovered S, M, and L genome segment sequences (1299, 453, and 412 nt in length, respectively) for 33 PUUV strains circulating in the Ardennes were deposited in GenBank under accession numbers MN241149–MN241247.

### 2.6. Molecular Signatures of Selection of PUUV Strains

Nucleotide sequence alignments were generated, and genetic distances (within and between viral populations), haplotype diversity, nucleotide diversity, Tajima’s *D*, and Fu and Li’s *D* and *F* tests were estimated with DnaSP v.6 [76]. Such tests distinguished between a neutral evolution and nonrandom evolutionary processes (i.e., directional selection or balancing selection). Additionally, genetic selection was evaluated on the Datamonkey Adaptive Evolution Server [77] using different codon-based maximum-likelihood (ML) methods: fixed-effects likelihood (FEL), single-likelihood ancestor counting (SLAC), mixed effects model of evolution (MEME), a codon-based Bayesian approach (fast unbiased approximate Bayesian (FUBAR)), a “branch-site” model (the adaptive branch-site random effects likelihood (aBSREL)) and a “gene-wide” model (branch-site unrestricted statistical test for episodic diversification (BUSTED)). The ML methods estimate the dN/dS rate at every codon in the alignment. FEL uses an ML approach to infer nonsynonymous (dN) and synonymous (dS) substitution rates on a per-site basis for a given coding alignment and corresponding phylogeny. SLAC uses a combination of ML and counting approaches to count the number of nonsynonymous changes per nonsynonymous site (dN) and tests whether it is significantly different from the number of synonymous changes per synonymous site (dS). Like FEL, SLAC assumes that the selection pressure for each site is constant along the entire phylogeny. MEME employs a mixed-effects ML approach aiming to detect sites evolving under positive selection or diversifying selection under a proportion of branches. FUBAR analyzes the coding sequence alignment to determine whether some sites have been subject to pervasive purifying or diversifying selection. aBSREL is a “branch-site” model that tests if positive selection has occurred on a proportion of branches. Lastly, BUSTED provides a gene-wide (not site-specific) test for positive selection by asking whether a gene has experienced positive selection at a minimum of one site on at least one branch. 

### 2.7. Phylogenetic Analysis of PUUV Strains

Phylogenetic analyses for the three genome segments were performed with both ML and Bayesian methods, implemented in RAxML Blackbox webserver [78] and BEAST v2.6.2 [79], respectively. The general time reversible (GTR) and gamma site model with invariant sites (GTR + G + I) model of evolution was used as determined as best fit by jModelTest v2.1.10 [80]. For the Bayesian analyses, a lognormal relaxed clock (allowing branch lengths to vary according to an uncorrelated lognormal distribution) was chosen, with a nonparametric and very flexible coalescent Bayesian skyline tree prior, allowing the population size to vary stochastically through time. These two complementary and nonredundant phylogenetic methods were implemented: RAxML for its rapid bootstrap analysis and BEAST 2 to back up topologies. The obtained trees were edited with FigTree v1.4.3 [81]. To focus on our study area, we conducted additional phylogenetic analyses using PUUV sequences coming from our sampling locations and three outgroups of the CE lineage (PUUV/Jura/Cg0510y27r/2010, Orleans_23, and NL.MG31.2007). These phylogenetic analyses were also done using ML approaches as described above.

### 2.8. Local Genetic Diversity

The genetic parameters estimated for each site were used as surrogates of demographic features for both PUUV and bank voles, including population size and migration [73]. Genetic diversity of PUUV was estimated at each sampling site by calculating the number of nucleotide and amino-acid substitutions for S, M, and L genome segments using DnaSP v.6 [76]. The allelic richness (A) and genetic diversity in bank voles were determined within each site by estimating observed (H_O_) and Nei’s unbiased expected (H_E_) heterozygosities as described before [73]. For each site, the local genetic variance (F_ST_) was estimated to evaluate the level of genetic isolation of each site with regard to all other ones. More details in genetic structure analyses can be found in our earlier study [73]. 

### 2.9. Large-Scale Genetic Structure

For these analyses, we used the three concatenated genome segments of PUUV (see Section 2.5 or Table 2 for the details of the portions of segments used) and 16 neutral microsatellite loci for bank voles. Similar genetic analyses for both PUUV and bank voles were implemented in GenAIEx 6.5 [82,83]. Hierarchical partitioning of genetic variation among localities and forest massifs was estimated using analyses of molecular variances (AMOVA). Such analysis was calculated using a haploid distance matrix for the calculation of Phi_PT_ for PUUV and a codominant allelic distance matrix for the calculation of F_ST_ for bank voles. AMOVA calculated estimates of variation among forest massifs, among localities within forest massifs, and among individuals within localities. The spatial genetic structures were investigated using principal coordinate analysis (PCoA), a multivariate technique that allows finding and plotting the major patterns within a multivariate data set. PCoA of the studied populations was carried out using the covariance-standardized matrix method in GenAlEx. Lastly, the isolation by distance (IBD) was calculated with a Mantel test based on 999 permutations to assess the significance of the regression between the pairwise genetic distance (Phi_PT_ for PUUV and F_ST_ for bank voles) and the logarithm of the geographic distance. 

### 2.10. Landscape Analyses

We investigated the effects of landscape structure on the genetic diversities using statistical logistic regressions performed with the RStudio platform v1.2.1335 [84], applying the packages MuMIn v1.43.6 [85] and nlme v3.1-140 [86]. Model selection was performed using Akaike’s information criterion (AIC) [87,88]. The model with the lowest AIC value was viewed as the most parsimonious, i.e., the model explaining the majority of variance with the fewest parameters [88]. The significance of each explanatory variable was tested using Wald tests based on *z*-values. Beforehand, we calculated the correlations between each variable of the landscape and the PUUV and bank vole datasets, in order to detect highly correlated variables which may have hampered GLM analyses because of multicollinearity [89]. Correlation matrices were calculated using the RStudio platform and the package corrplot v0.84 [90].

## 3. Results

In order to understand the potential contribution of landscape genetics to host-specific virus distribution and, subsequently, virus genetic diversity, we evaluated the correlation of genetic structure and diversity of PUUV with that of its specific host, the bank vole. For a broader understanding, we incorporated the results from our previous study, where landscape genetics were used to investigate how the population dynamics of the bank vole vary with forest fragmentation and influence PUUV epidemiology [73]. Here, we genetically analyzed the 33 infecting PUUV and considered the genetic diversity of 310 bank voles in 10 localities of Northeast France, in the Ardennes department. Out of the 33 analyzed PUUV partial genomes, all but two were recovered from voles trapped in 2008, and the other two were recovered from bank voles circulating in the locality of Boult-aux-Bois during 2010 (Figure 1). Each of the studied localities was characterized by its landscape features (Table 1). Below, we explain the outcome of our analyses.

### 3.1. Detection and Genetic Diversity of PUUV and Bank Voles from the French Ardennes

We sampled 310 bank voles at 10 different localities with geographical distances of at least 3.2 km. PUUV-specific antibodies were detected by IFAT assay in 38 voles; accordingly, the PUUV prevalence for the studied bank vole population was 12.3%. PUUV could not be recovered by RT-PCR from five seropositive voles; four of those were identified as young animals at the sampling time, suggesting that the detected PUUV antibodies could correspond to maternal antibodies. From the 33 remaining samples, PUUV genome sequences were examined; pairwise analysis of 1299 nucleotides (1–1299 nt) of the nucleocapsid-encoding S segment, 453 nucleotides (2180–2632 nt) of the M segment, and 411 nucleotides (577–987 nt) of the L segment revealed that PUUV strains from the French Ardennes exhibited a high sequence variation (S = 0–14.2%; M = 0–13.9%; L = 0–17.3%) (see Table 2) compared to other PUUV datasets published earlier at similar geographical scales [47,49,52]. Within sampling sites, the diversity of the analyzed genomes was much lower, ranging from 0% to 6.8%. In two of the sampling localities (Sauville and Briquenay), PUUV was not detected. Nevertheless, the genetic diversity of strains from the Woiries locality was higher than from the others (Table 2).

Along these lines, the chromatogram of the L genome segment of the Woiries_58 strain showed double peaks at certain nucleotide sites. These double peaks were distributed not randomly along the sequence but appeared exclusively at nucleotide positions that genetically discriminate strains from the different studied localities (Appendix A). 

Overall, there was a high diversity of virus sequences with 30 different types for the S segment, 14 for the M segment, 12 for the L segment, and 31 when all three segments were taken into account (concatenated segments). Only a few amino-acid substitutions were observed for the three genome segments (Table 2). Moreover, evolutionary neutrality analyses displayed nonsignificant values of Tajima’s D, as well as Fu and Li’s D and F tests (Table 3). The only significantly positive values were obtained with Fu and Li’s D and F tests for the L genome segment of the Ardennes strains (Table 3). 

Furthermore, the evaluation of genetic selection of the Ardennes strains with FEL, SLAC, and FUBAR revealed abundant sites under negative (purifying) selection (Appendix A), whilst FEL, SLAC, MEME, aBSREL, and BUSTED found no evidence for positive (diversifying) selection for any of the three genome segments of PUUV from the Ardennes (Appendix A). 

Only FUBAR found evidence for codon-positive selection in the S genome segment (at site 265). Among bank voles, both heterozygosity (H0, HE) and allelic richness (A) strongly differed between sampling sites (Appendix A); they were higher in the forested habitats than in hedge areas. Analyses of F-Statistics (FIS, FST) corroborated a stronger genetic isolation in the hedge areas than in forest habitats (Appendix A). See our previous study [73] for further details on bank vole genetics analyses.

### 3.2. Phylogenetic Analysis of PUUV Strains from the French Ardennes

Nearly identical tree topologies were obtained using the ML and the Bayesian analyses for the S and partial M and L segment sequences. Phylogenies showed that all the PUUV strains recovered from the French Ardennes between 2008 and 2010 clustered together and belong to the CE lineage (Appendix A), including previously described closely related PUUV strains coming from the northern part of France, Belgium, Germany, and the Netherlands, namely Momignies, Montbliart, CG14444, CG1445, Couvin, Thuin, Cg-Erft, CG13891, NL_Mg2_2008, NL_Mg502_2008, and NL_Mg591_2008. 

Furthermore, strains from the French Ardennes appeared to be less related to strains recovered from the central and eastern parts of France (Orleans, Alsace, and Jura), Germany, and the eastern part of the Netherlands (Appendix A). Three main clusters could be observed among the French Ardennes strains corresponding to the geographical distribution of the sampled localities in the studied forest massifs, congregating as follows (Figure 2 and Appendix A): strains from Boult-aux-Bois and Croix-aux-Bois (south forest massif, in orange in Figure 1), strains from Elan and Cassine (central forest massif, in blue in Figure 1), and strains from Hargnies, Renwez, Cliron, and Woiries (north forest massif, in green in Figure 1). Similarly, for the three genome segments, all Ardennes strains grouped together following the geographical distribution of the studied localities, except for the strains from the Woiries locality. As mentioned above, genetic variability in Woiries was higher than that of the other Ardennes localities (Table 2), and this was also reflected by the phylogenies (Figure 2 and Appendix A), showing two of the strains (Woiries_44 and Woiries_63) clustering with the strains of the north forest massif (Hargnies, Renwez, and Cliron localities), whereas the strain Woiries_58 did not always group with the same phylogenetic cluster, aligning with different groups depending on the analyzed genome segment. Its S and M genome segments did not cluster with any Ardennes genotype cluster, although its L genome segment was similar to the Hargnies genotypes. We reiterate that the chromatogram of the Woiries_58 strain for the L genome segment showed double peaks in the nucleotide sites for locality group discrimination. It should also be noted that, whilst Woiries_44 presented all three segments similar to the Renwez genotypes (S-Renwez/M-Renwez/L-Renwez), the S and M segments of Woiries_63 were comparable to the Renwez genotypes, whilst the L segment was closer to the Hargnies genotypes (S-Renwez/M-Renwez/L-Hargnies) (Figure 2 and Appendix A). Generally, phylogenetic analyses of PUUV Ardennes strains disclosed that strains generally gathered with the closest forest massif neighbor. However, we detected an exception with the PUUV strains from Hargnies and Cliron sites (Figure 2 and Appendix A).

### 3.3. Spatial Genetic Structure of PUUV and Bank Voles from the French Ardennes

Neutral microsatellite loci were used to study the genetic diversity of bank voles and monitor their local population dynamics. The bank vole and PUUV populations were analyzed for genetic variation among and within the clusters using AMOVA. Such analyses (Table 4) showed that the genetic variance of bank voles (1%) was weakly structured between the three studied forest massifs, whilst the genetic variance of PUUV between forest massifs was significantly higher (65%). 

Additionally, a significant pattern of isolation by distance (IBD) was detected for PUUV (*r* = 0.27; *p* = 0.05). In contrast, IBD was not observed in our studied bank voles (*r* = 0.01; *p* = 0.24) (Figure 3). 

Furthermore, PCoA were used to decipher the spatial genetic structure of our samples. These analyses revealed that PUUV strains from the same forest massif assembled together; this grouping was remarkable since PUUV strains belonging to locations of the same forest massif were much closer to each other than those from localities belonging to other forest massifs (Figure 4). As represented in Figure 4, the first two axes of the PCoA of PUUV exhibited a high proportion (81%) of the genetic diversity (axis 1: 54.4%; axis 2: 26.6%). In a different manner, the spatial genetic structure of bank voles was fuzzier; voles from two of the localities of different forest massifs were closer to each other than those from localities from the same forest massif. Overall, the vole’s genetic structuration was sparse as indicated by lower numbers in axis 1 and axis 2 of the bank vole’s PCoA. These two axes explained 46.9% of the genetic variation (axis 1: 28%; axis 2: 18.9%) (Figure 4). 

### 3.4. Effect of Landscape Features and Bank Vole Population Dynamics on PUUV Prevalence and Diversity

We independently calculated a correlation matrix for landscape, bank vole, and PUUV variables to detect the values that were highly correlated, aiming to preserve only the weakly or not correlated variables of our datasets. Accordingly, we conserved the following variables: the proportion of forested habitat (i.e., Forest), the forest contiguity (i.e., Contig), and the forest shape (i.e., Shape) variables of the landscape dataset (Table 1), the allelic richness (A) variable of the bank vole’s dataset (Appendix A), and the nucleotide diversity (i.e., NtDiv) and amino-acid diversity (i.e., AaDiv) variables of the PUUV dataset (Table 2). Bank vole diversity was positively correlated with landscape features; the best model based on AIC values only included the Forest variable (AIC: 17.7; *z*-value: 2.909, Pr(>|z|): 0.01). PUUV prevalence was also highly correlated with landscape features; notably, the best model based on AIC values included both the Forest and the Contig variables (AIC: 52.3). Along these lines, PUUV prevalence was positively correlated with the Forest variable (AIC: 52.50; *z*-value: 3.049, Pr(>|z|): 0.002) and with the Contig variable (AIC: 52.67; *z*-value: 3.012, Pr(>|z|): 0.002). PUUV prevalence was also remarkably correlated with the genetic diversity of bank vole populations (AIC: 46,67; *z*-value: 3.796, Pr(>|z|): 0.0001). PUUV nucleotide diversity was correlated with landscape features; notably, the best model based on AIC values included both the Forest and the Contig variables (AIC: 31.8), with PUUV nucleotide diversity being positively correlated with the Forest variable (AIC: 32.033; *z*-value: 3.289, Pr(>|z|): 0.01) and with the Contig variable (AIC: 35.132; *z*-value: 2.409, Pr(>|z|): 0.04). On the other hand, PUUV amino-acid diversity was not significantly correlated with eeither of the landscape variables (Forest: *z*-value: 1.558, Pr(>|z|): 0.15; Contig: *z*-value: 2.127, Pr(>|z|): 0.06; Shape: *z*-value: 0.981, Pr(>|z|): 0.35). Lastly, PUUV diversity (neither NtDiv nor AaDiv) was not correlated with the bank vole genetic diversity (*z*-value: 0.858, Pr(>|z|): 0.416; and *z*-value: 1.727, Pr(>|z|): 0.122; respectively) (Table 5).

## 4. Discussion

Parasites, especially specific microparasites such as viruses, strongly depend on their host. In turn, hosts are steadily influenced by landscape features, affecting population distribution, dynamics, density, and abundance of susceptible individuals. Thus, environmental factors may often be the ultimate drivers modeling hosts and, consequently, parasites [15]. In nature, a large group of parasites are mainly transmitted by close contact (i.e., they depend on their hosts/vectors for dispersal), and their genetic layout should correlate with the environmental features which have a strong influence on host/vector dynamics and population structure [91]. This is especially likely when parasites have a low impact on their host fitness, as is the case for most zoonotic microparasites. The structure of a landscape, such as the size range and spatial distribution of habitat patches of host populations, may be a critical determinant of the parasite genetic structure. The genetic pattern of a particular emerging parasite species will largely depend on the host dispersal and the overall host population dynamics [15]. Hence, host specificity is expected to have a strong impact on the distribution, the prevalence, and the genetic diversity of their parasites [92]. In particular, habitat preferences and population dynamics of the host should shape the local prevalence and the genetic diversity of its host-specific parasite. Such assumptions are rarely documented in nature. Understanding the landscape genetic patterns may help to unravel the complex history of host–parasite coevolution [93]. Availability of such information is critical from a disease management perspective since it could be used to develop local strategies for surveillance and control of potential zoonotic outbreaks [94]. This study illustrates the potential contribution of landscape genetics for understanding the link between a host-specific virus distribution and its genetic diversity. Here, we show how the genetic structure and genetic diversity of a virus and its specific host, driven by several landscape parameters, are correlated. 

### 4.1. Genetic Diversity of PUUV and Bank Voles from the French Ardennes

From the inspection of the genetic diversity of PUUV from the French Ardennes, we can conclude that nucleotide diversity is comparable with that of PUUV strains belonging to other known PUUV lineages (Table 3). Nonetheless, it is appreciable that the genetic diversity of the PUUV Central European (CE) lineage is greater than that of other lineages (Table 6); these data might be a consequence of the analysis of a greater number of strains (i.e., larger number of genetic sequences available for this lineage up to now) that come from a broad geographical extension. 

Furthermore, it should be noted that the genetic diversity of strains from Woiries was higher than that of strains from other Ardennes localities (Table 2); this was due to the presence of an eccentric strain, i.e., Woiries_58, which is certainly genetically disparate to its other Ardennes relatives. Furthermore, the haplotype diversity of the Ardennes PUUV strains was comparable to strains from the CE lineage and, in general, to all other PUUV strains (Table 3). Notwithstanding, nonsignificant results for Tajima’s *D* and Fu and Li’s *D* and *F* tests suggest that the null hypothesis of neutral evolution cannot be rejected. Additionally, the significantly positive value obtained with Fu and Li’s *D* test for the L genome segment of the Ardennes strains suggests a decrease in the population size and/or balancing selection (Table 3). Nevertheless, this speculation should be taken with caution since PUUV sequences across the sampling locations probably do not form a single mixed population. This was also the case when analyzing PUUV circulating only in 2008 (31 individuals coming from all locations but the Boult-aux-Bois site). Furthermore, as already documented earlier for PUUV [50,51,52,68], the evaluation of genetic selection of the Ardennes strains revealed abundant sites under negative selection (Appendix A) and only evidence for positive selection for the S genome segment of PUUV from the Ardennes (Appendix A). This strong negative selection indicates that those new deleterious or less fit variants were removed from the population. As suggested before in Razzauti et al. 2013 [52], the absence of signatures for positive selection indicates that there is no increment of variants that would confer a fitness advantage relative to the rest of the population or increase of its genetic diversity.

On the other hand, heterozygosity and allelic richness in bank voles strongly differed across the sampling locations (see in [73] and Appendix A), with both being higher in forested habitats than in hedge areas. As expected, the genetic isolation in bank voles was more evident in hedge areas than in forested habitats.

### 4.2. Phylogenetic Analysis of PUUV Strains from the French Ardennes

Our analyses revealed that PUUV genetic diversity in Woiries was higher than that in the other Ardennes localities; this was, in particular, caused because the strain Woiries_58 switched genetic clusters depending on the analyzed genome segment (Figure 2 and Appendix A). More interestingly, we noticed that, whilst the strain Woiries_44 presented all three segments similar to the Renwez genotypes (S-Renwez/M-Renwez/L-Renwez), the strain Woiries_63 displayed the S and M segments like the Renwez-genotype but the L segment belonging to the Hargnies genotype (S-Renwez/M-Renwez/L-Hargnies) (Figure 2 and Appendix A). Unlike Woiries_58, no double peaks were observed in the chromatograms of Woiries_44 and Woiries_63 (or any other here studied strain). Hence, we can speculate that a genome reassortment event could have happened in strains from the Woiries location. These particularities could also suggest the Woiries locality as a contact zone for the PUUV strains circulating in Renwez and Hargnies areas of the studied forest massifs (Figure 1). Furthermore, we observed a phylogenetic discordance among the PUUV strains from the Ardennes; whilst all PUUV strains generally gathered with strains of their closest neighbor forest massif, strains from Hargnies and Cliron localities seemed to be an exception (Figure 2 and Appendix A). This controversy could be explained by our previous study [73], where evidence for bank vole migration was detected from Hargnies (forested area) to Cliron (hedge area) (see Figure 1). This was also reflected in PUUV strains; whilst the Cliron hedge site was farther away of Hargnies locality than other forested sites of the same forest massif (i.e., Renwez and Woiries), strains from Cliron genetically clustered with strains from Hargnies. Furthermore, we earlier identified [73] migrant PUUV-infected voles in Boult-aux-Bois (hedge area); indeed, strains from this site clustered together with strains from Croix-aux-Bois (forested area), perhaps indicating that those migrant voles carried PUUV strains from the hedge site to the closest forested locality of Croix-aux-Bois. In Guivier et al., 2011 [73], it was also suggested that bank voles from these two hedge sites, Cliron and Boult-aux-Bois (Figure 1), were genetically isolated and that they accommodated PUUV-seropositive immigrant bank voles from forested areas. Upon the presence of PUUV genetically close strains, we speculate that these immigrants could have carried the virus with them and spread such variants to the hedge areas. This would explain the likelihood of PUUV propagation through vole dispersal between different areas and landscapes. 

### 4.3. Spatial Genetic Structure of PUUV and Bank Voles from the French Ardennes

Through the analyses of neutral microsatellites loci, we studied the genetic diversity of bank voles and monitored their local population dynamics. The results of PCoA evidenced that landscape features might shape bank vole populations and this, in turn, the prevalence of PUUV. The results also illustrated that PUUV was strongly structured in space in a hierarchical manner; from the large spatial scale to the fine spatial scale. Contrarily, bank voles were structured in a lower degree at large spatial scale, and most of the genetic variation was observed at a finer scale within populations. Hence, bank vole populations in forested areas were genetically homogeneous, whilst fragmented landscapes showed gene flow/genetic drift, revealing an asymmetric gene flow from forest to hedge populations. Nevertheless, some signatures of gene flow/genetic drift and genetic isolation in bank voles from locations with low forest coverage and contiguity (i.e., fragmented habitats) indicated an asymmetric gene flow from forest to hedge populations. However, the integrity of bank vole genetic populations was not affected by stochastic events of local extinction/recolonization.

### 4.4. Effect of Landscape Features and Bank Vole Population Dynamics on PUUV Prevalence and Diversity

Our results here and before [73] clearly show that bank vole diversity is regulated by landscape features. In a similar manner, PUUV prevalence also seems to be indirectly shaped by landscape features. Data analyses also revealed that PUUV prevalence is remarkably correlated with bank vole genetic diversity. In this study, a more genetically homogenic host population led to more virus being found circulating in that population. Could this be because of the virus adaptation to that host genotype or simply because the host individuals are close-circulating and, thus, the virus has a better chance to be spread throughout the population? These interrogations merit further investigations. Nonetheless, no genetic structure was detected at the level of the microsatellite markers, suggesting that the virus might not overcome an adaptation to a host genotype. Moreover, our investigation showed that the nucleotide diversity of PUUV seems to be also influenced by the landscape features, meaning that the evolution of the virus might be indirectly modulated by the environment. On the other hand, neither nucleotide nor amino-acid diversity of PUUV was correlated with the bank vole diversity, showing a virus rate of evolution different and independent to that of the host.

## 5. Conclusions

Since the start of PUUV studies, it was evident that host abundance is a modulator of the microparasite prevalence. However, there are not many reports where bank vole diversity or landscape features were considered to predict the virus prevalence and/or emergence. We studied the genetic diversity and distribution of PUUV strains in its specific host population, incorporating the environmental features affecting the host–virus tandem into the analysis. Our data strongly support the hypothesis that the bank voles display an elevated gene flow and large population sizes over broad forested areas. Nevertheless, bank vole populations showed signs of genetic drift and genetic isolation in the studied locations with low coverage and contiguity of forest habitats. Moreover, bank vole immigration from large continuous populations into small isolated ones could be an essential process for the occurrence and transmission of PUUV in fragmented landscapes and, consecutively, the expansion or the extinction of the virus. According to our results, PUUV prevalence is also positively correlating with the coverage and the contiguity of forested habitats. On the other hand, the genetic diversity of PUUV is much more structured in space, suggesting higher genetic drift and lower gene flow of PUUV between forest massifs than that of its host. Hence, we assume that vole dispersion is not sufficient to homogenize PUUV genotypes in large forested areas. As expected, we found evidence for purifying selection for the Ardennes PUUV and hypothesize that demographic stochasticity (genetic drift and extinction/recolonization events) is, more likely, the main evolutionary process in action. Lastly, at the spatial scale of our study (80 km), both bank vole and PUUV populations are mainly shaped by the coverage and the contiguity of forest habitats. However, bank vole populations would be less affected by stochastic events of local extinction/recolonization and genetic drift than PUUV, its specific parasite. These contrasting patterns of microevolution might have important consequences for understanding geographic distribution and epidemiology of PUUV. Altogether, our study emphasizes that landscape genetics have a central role in zoonotic diseases studies.

## Figures and Tables

**Figure 1 microorganisms-09-01516-f001:**
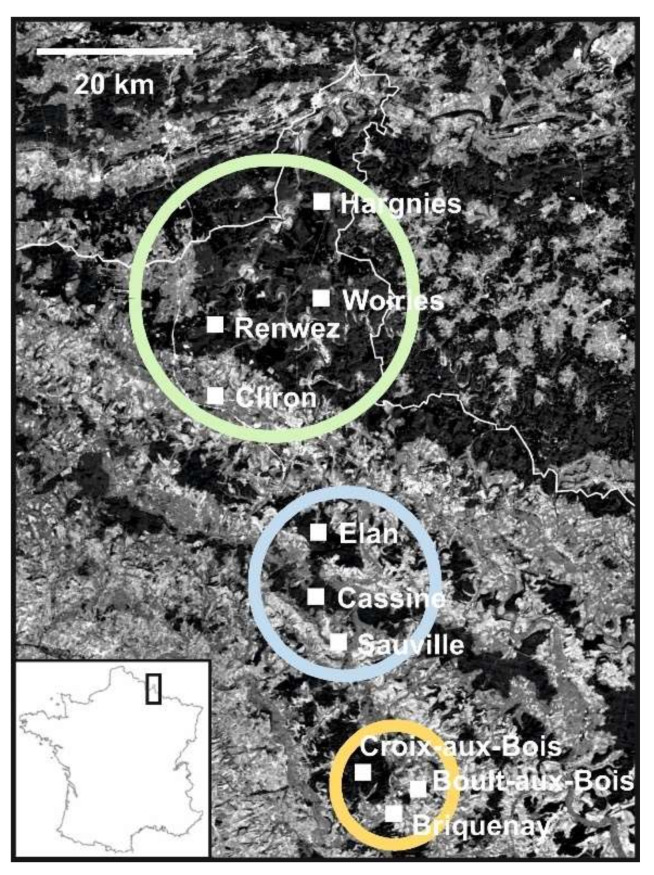
Map of the trapping sites. The map represents the Ardennes in the north of France, showing the location of the 10 trapping sites. Dark areas correspond to forest habitats. Green, blue, and orange circles indicate the three different forest massifs explored in the study.

**Figure 2 microorganisms-09-01516-f002:**
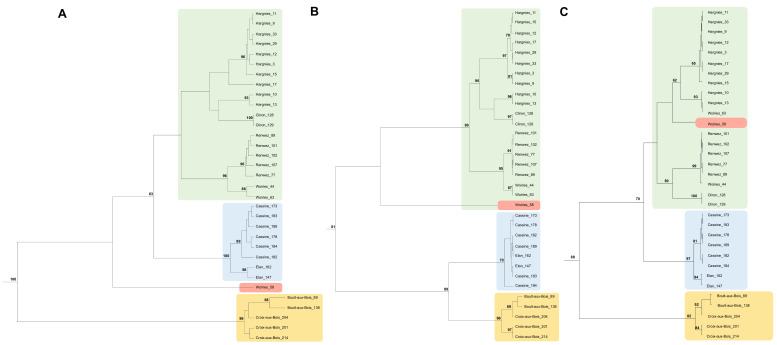
Phylogenetic analyses of *Puumala* hantavirus strains of the French Ardennes. Maximum-likelihood and Bayesian trees calculated for the S, M, and L segment sequences of PUUV: (**A**) S sequences (43–1341 nt); (**B**) partial M sequences (2180–2632 nt); (**C**) partial L sequences (577–987 nt). Three outgroup strains (PUUV/Jura/Cg0510y27r/2010, Orleans23, and NL.MG31.2007) belonging to other sub-lineages of the CE lineage are not represented in the figure for its easy representation. Maximum clade credibility trees are presented with nonparametric bootstrap support values (>60%) shown on nodes. Colors represent the three forest massifs (green: north, blue: central, and orange: south, as represented in Figure 1). Since Woiries_58 clustered differently in the three trees, it is represented in red for contrast.

**Figure 3 microorganisms-09-01516-f003:**
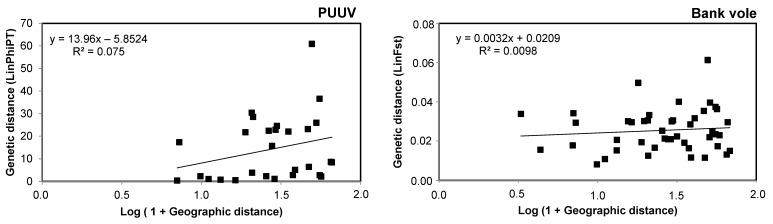
Isolation by distance (IBD) for *Puumala* hantavirus (PUUV) and bank voles. The graphics represent the population pairwise genetic distance plotted against the geographic distance between paired populations. On the left side, the results of the Mantel test for the analyzed sequences of PUUV (concatenated S, M, and L genome segments; 2163 nt) and, on the right side, the results of Mantel test for the 16 microsatellite loci.

**Figure 4 microorganisms-09-01516-f004:**
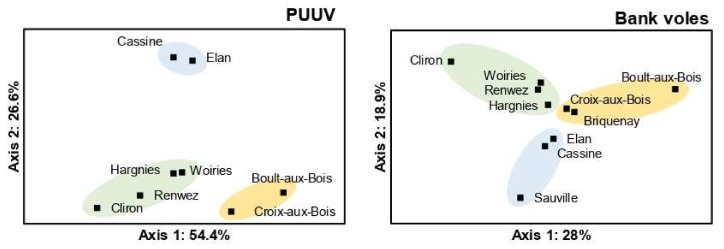
Principal coordinates analyses (PCoAa) for *Puumala* hantavirus and bank voles. Colors represent the three forest massifs (green: north, blue: central, and orange: south, as represented in Figure 1).

**Table 1 microorganisms-09-01516-t001:** Geographical coordinates and landscape variables of the different sampling locations.

Sites	Longitude	Latitude	Proportion of Forested Habitat (%)	Forest Contiguity Index	Shape Index	Edge Density (m/km^2^)	No. of Patches	Distance to Next Patch (m)
Hargnies	4.7950	49.9961	0.838	0.93	37.3	883.7	14	200
Woiries	4.7664	49.9037	0.861	0.93	37.3	936.4	9	200
Renwez	4.6113	49.8590	0.632	0.93	37.3	14484	10	200
Cliron	4.6204	49.8032	0.022	0.85	19.4	169.7	2	956
Elan	4.7674	49.6536	0.585	0.71	4.1	1203.3	3	200
Cassine	4.7945	49.5752	0.421	0.84	3.5	1226.3	3	208
Sauville	4.8003	49.5454	0.252	0.82	3.0	901.9	4	411
Croix-aux-Bois	4.8385	49.4166	0.596	0.82	2.8	900.2	9	355
Boult-aux-Bois	4.9215	49.4136	0.187	0.76	2.5	1311.8	8	445
Briquenay	4.9109	49.3937	0.169	0.78	2.7	904.5	7	493

**Table 2 microorganisms-09-01516-t002:** Genetic diversity of *Puumala* virus (PUUV) strains from les Ardennes.

Genome Segment		No. of Strains	S (43–1341 nt)	M (2180–2632 nt)	L (577–987 nt)
Analyzed length	Nucleotides (nt)	33	1299	453	411
	Amino acids (aa)	33	433	151	137
No. of substitutions	Nt polymorphic sites (total no. of mutations)	33	178 (185)	62 (63)	69 (71)
	Nt singleton variable sites	33	29	10	2
	Nt parsimony informative sites	33	149	52	67
	Aa variable sites	33	11	4	5
Genetic diversity (%)	Nucleotides (nt)	33	14.2	13.9	17.3
	Amino acids (aa)	33	2.5	2.7	2.7
No. of nt substitutions (%)	Hargnies	10	29 (2.2)	11 (2.4)	13 (3.2)
	Woiries	3	60 (4.6)	25 (5.5)	28 (6.8)
	Renwez	5	10 (0.8)	0	0
	Cliron	2	0	0	0
	Elan	2	4 (0.3)	1 (0.2)	1 (0.2)
	Cassine	6	10 (0.8)	2 (0.4)	0
	Croix-aux-Bois	3	8 (0.6)	4 (0.9)	3 (0.7)
	Boult-aux-Bois	2	4 (0.3)	1 (0.2)	0
No. of aa substitutions (%)	Hargnies	10	3 (0.7)	0	0
	Woiries	3	1 (0.2)	0	2 (1.5)
	Renwez	5	2 (0.5)	0	0
	Cliron	2	0	0	0
	Elan	2	0	0	0
	Cassine	6	1 (0.2)	0	0
	Croix-aux-Bois	3	0	1 (0.7)	1 (0.7)
	Boult-aux-Bois	2	0	0	0

**Table 3 microorganisms-09-01516-t003:** Neutrality tests, haplotype, and nucleotide diversity analyses of *Puumala* virus (PUUV) from the Ardennes.

Segment	Strains Group	*n*	π	h	Hd	Tajima’s *D*	Fu and Li’s *D*	Fu and Li’s *F*
	Ardennes	33	0.036	30	0.994	0.10	0.64	0.50
S	CE	99	0.098	89	0.997	0.53	0.62	0.70
	All PUUV	189	0.138	178	0.999	0.46	−0.04	0.25
	Ardennes	33	0.043	14	0.922	0.99	0.59	0.86
M	CE	63	0.096	29	0.960	0.75	1.12	1.17
	All PUUV	117	0.147	76	0.987	0.62	0.88	0.91
	Ardennes	33	0.051	12	0.898	0.75	1.70 *(*p* < 0.02)	1.63 *( *p* < 0.05)
L	CE	52	0.106	26	0.953	0.49	0.68	0.73
	All PUUV	88	0.160	61	0.984	0.31	0.67	0.62

S = small PUUV genome segment; M = medium PUUV genome segment; L = large PUUV genome segment; Ardennes = strains recovered from les Ardennes for this study; CE = strains of the Central European lineage; all PUUV = *Puumala* virus strains known to the present; *n* = number of studied sequences; π = nucleotide diversity; h = number of haplotypes; Hd = haplotype diversity; * statistically significant values.

**Table 4 microorganisms-09-01516-t004:** Analysis of Molecular Variance (AMOVA).

	Source	df	SS	MS	Est. var.	%	Stat.	*p* (Rand ≥ Data)
PUUV	Among forest massifs	2	896.9	448	41.3	65	Phi_RT_ = 0.647	0.001
	Among localities	5	313.1	62.6	14.8	23	Phi_PR_ = 0.661	0.001
	Within localities	25	190.9	7.6	7.6	12	Phi_PR_ = 0.661	0.001
	Total	32	1400		63.8	100	Phi_PT_ = 0.880	0.001
Bank voles	Among forest massifs	2	49.8	24.9	0.04	1	F_RT_ = 0.006	0.001
	Among localities	7	111.9	16	0.1	2	F_SR_ = 0.020	0.001
	Within localities	300	2199.2	7.3	0.3	4	F_SR_ = 0.020	0.001
	Total	619	4435.8		7.2	100	F_ST_ = 0.026	0.001

For bank voles: input as codominant allelic distance matrix for calculation of F_ST_, (within individual analysis suppressed). For PUUV: input as haploid distance matrix for calculation of Phi_PT_. df = degrees of freedom; SS = sum of squares; MS = mean squares; Est. var. = Estimated variance; % = percentage of genetic variation; Stat. = statistic used to estimate structure; p(rand≥data) = probability of obtaining a F/Phi value of X or greater if there were no differences between the groups. F-statistics = the statistically expected level of heterozygosity in a population; Phi-statistics = analogue of F statistic, describing the statistically expected level of variability in a population.

**Table 5 microorganisms-09-01516-t005:** Summary of the general linear model (GLM) analyses.

	Source	Landscape			Vole
		% Forest	Contiguity	Shape	Genetic Diversity
Voles	Genetic diversity	+ **	NS	NS	
	PUUV prevalence	+ **	+ **	NS	+ ***
PUUV	Nucleotide diversity (*NtDiv*)	+ **	+ *	NS	NS
	Amino-acid diversity (*AaDiv*)	NS	NS	NS	NS

(+) = positively correlated; NS = not significant; * statistically significant values: (*) *p* < 0.05; (**) *p* < 0.01; (***) *p* < 0.001.

**Table 6 microorganisms-09-01516-t006:** Genetic variation of known Puumala virus (PUUV) strains.

Genome Segment	PUUV Lineages	No. of Strains	No. of nt Substitutions (Variable Sites)	Genetic Diversity (%)	aa Substitutions (%)
S (43–1341 nt; 1299 nt, 433 aa)	All known PUUV	189	913 (584)	70.3	73 (16.9)
	Danish (DAN)	3	87 (85)	6.7	17 (3.9)
	Latvian (LAT)	5	178 (174)	13.7	1 (0.2)
	North-Scandinavian (N-SCA)	23	335 (294)	25.8	20 (4.6)
	South-Scandinavian (S-SCA)	8	261 (251)	20.1	17 (3.9)
	Russian (RUS)	16	364 (319)	28.0	29 (6.7)
	Finnish (FIN)	27	429 (361)	33.0	52 (12.0)
	Alpe-Adrian (ALAD)	8	138 (133)	10.6	13 (3.0)
	Central European (CE)	99	567 (435)	43.6	31 (7.2)
M (2180–2632 nt; 453 nt, 151 aa)	All knwon PUUV	117	258 (167)	57.0	13 (8.6)
	Danish (DAN)	3	21 (21)	4.6	3 (2.0)
	Latvian (LAT)	1	-	-	-
	North-Scandinavian (N-SCA)	8	128 (113)	31.8	9 (6.0)
	South-Scandinavian (S-SCA)	4	81 (77)	17.9	3 (2.0)
	Russian (RUS)	11	145 (121)	32.0	8 (5.3)
	Finnish (FIN)	15	144 (124)	31.8	8 (5.3)
	Alpe-Adrian (ALAD)	12	80 (71)	17.7	6 (4.0)
	Central European (CE)	63	146 (127)	32.2	12 (7.9)
L (577–987 nt; 411 nt, 137 aa)	All known PUUV	88	304 (202)	74.0	21 (15.3)
	Danish (DAN)	3	32 (32)	7.8	4 (2.9)
	Latvian (LAT)	4	91 (83)	22.1	8 (5.8)
	North-Scandinavian (N-SCA)	2	61 (61)	14.8	10 (7.3)
	South-Scandinavian (S-SCA)	0	-	-	-
	Russian (RUS)	7	122 (108)	29.7	12 (8.8)
	Finnish (FIN)	17	105 (95)	25.5	8 (5.8)
	Alpe-Adrian (ALAD)	3	21 (21)	7.8	0
	Central European (CE)	52	173 (140)	42.1	15 (10.9)
	North-Scandinavian (N-SCA)	23	335 (294)	25.8	20 (4.6)
	South-Scandinavian (S-SCA)	8	261 (251)	20.1	17 (3.9)
	Alpe-Adrian (ALAD)	8	138 (133)	10.6	13 (3.0)
	Central European (CE)	99	567 (435)	43.6	31 (7.2)

## Data Availability

The Ardennes PUUV genetic sequences are available in Genbank under accessions numbers MN241149–MN241247.

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
