# Peer review of "Impact of Landscape on Host–Parasite Genetic Diversity and Distribution Using the Puumala orthohantavirus–Bank Vole System"

_microorganisms, 2021, doi:10.3390/microorganisms9071516_

Round 1
Reviewer 1 Report
The manuscript entitled ’Impact of landscape on host-parasite genetic diversity and distribution using the Puumala orthohantavirus-bank vole system’ by Razzauti M et al. described the micro-evolution of Puumala virus (PUUV) carried by bank voles in the Northeast of France. Furthermore, authors would like to apply ecological data together with genetic data from host and virus, trying to answer the question if a certain landscape has impact on the genetic diversity of host-parasite system, which is a very important research topic regarding to the emerging viral zoonotic diseases. I have only one concern on the samples size (310 bank voles collected from 2008 and 2010), and ‘prevalence’ of PUUV (Table S2) in such a small samples size. Using such dataset during the GLM analysis would be very biased. However, maybe using sample with bootstrapping could increase the accuracy for such analysis hopefully.
Besides, there are some small points need to be clarified.
- Line 134-144, add samples information: how many samples have been collected and the years.
- Line 164-196, confusion on the definition of PUUV prevalence, and it was calculated using antibody positive samples or PCR positive samples?
- Figure 3, mantel test: why author using concatenated sequences, while in the phylogenetic trees (Figure 2 and S2), we already observed the reassortment events.
Reviewer 2 Report
The Authors reported about the impact of landscape on host-parasite genetic diversity and distribution. Studies that help to predict pathogen emergence and distribution are mandatory. I found the study of interest, well written and scientifically sound. I suggest a language revision by an English native-speaker.
Before acceptance I have some minor comments:
line 28: remove the full stop after reference 1
line 31: environmental changes of hosts? what do you mean?
line 55: delete the between is and more
lines 43-57: lots of repetition about this landcscape genetics perspective. try to condense.
lines 80-82: the sentence sounds strange
line 100: change virus population with virus itself
line 103: delete population and leave bank voles. deleta in this population after PUUV
lines 111-113: in this sentence the terms 'landscape features', 'land-surface attributes' , 'land cover' and 'land use' are repeated. Use one and simplify.
line 122: previous
line 137: voles' data
Figure 1: I don't like to see the forest areas fluctuating on a white background. Could the authors place them on a geographical context? Plus the green habitat is cut at its left and right appendixes.
line 161: suits well
line 165: not clear how the blood was collected and how it was measured to the cited quantity if it was places on a paper-strip.
line 280 delete have before evaluated
line 282: double 'the', delete one
line 296: PUUV could not be recovered by RT-PCR from five seropositive voles;
lines 307-308: the genetic diversity of strains from the Woiries locality was higher than from the others (Table 2).
line 342: clustered
line 349: Eastern
line 367: noted
line 457: on their host fitness
line 473: correlated
line 556: bank vole genetic diversity
line 561 the question 'could this be because the virus adaptation to that host-genotype.....' What? you mean 'because OF the virus adaptation? Also the second part of the question implies genetic is not involved.
line 569: delete 'not' before correlated, there's a double denial in the sentence.
Figure S2: the sentence 'A maximum clade credibility trees are shown as non-parametric bootstrap percentages (>
70%) are given at each node.' makes no sense grammatically.
